# Genetic influence is linked to cortical morphology in category-selective areas of visual cortex

Nooshin Abbasi[1,2], John Duncan[3,4] & Reza Rajimehr [3,5,6]*

Human visual cortex contains discrete areas that respond selectively to specific object categories such as faces, bodies, and places. A long-standing question is whether these areas are shaped by genetic or environmental factors. To address this question, here we analyzed functional MRI data from an unprecedented number ($n = 424$) of monozygotic (MZ) and dizygotic (DZ) twins. Category-selective maps were more identical in MZ than DZ twins. Within each category-selective area, distinct subregions showed significant genetic influence. Structural MRI analysis revealed that the 'genetic voxels' were predominantly located in regions with higher cortical curvature (gyral crowns in face areas and sulcal fundi in place areas). Moreover, we found that cortex was thicker and more myelinated in genetic voxels of face areas, while it was thinner and less myelinated in genetic voxels of place areas. This double dissociation suggests a differential development of face and place areas in cerebral cortex.

[1] McConnell Brain Imaging Centre, Montreal Neurological Institute, McGill University, Montreal, QC, Canada. [2] Interdisciplinary Neuroscience Research Program, Students' Scientific Research Center, Tehran University of Medical Sciences, Tehran, Iran. [3] MRC Cognition and Brain Sciences Unit, University of Cambridge, Cambridge, UK. [4] Department of Experimental Psychology, University of Oxford, Oxford, UK. [5] Cognitive Systems Lab, School of Electrical and Computer Engineering, College of Engineering, University of Tehran, Tehran, Iran. [6] School of Cognitive Sciences, Institute for Research in Fundamental Sciences (IPM), Tehran, Iran. *email: reza.rajimehr@mrc-cbu.cam.ac.uk

Visual object categorization is a fundamental cognitive process in human and nonhuman primates, which is thought to be mediated by clusters of neurons ("modules") within macroscopic regions in occipito-temporal and occipito-parietal cortex[1]. These category-selective areas are often named based on their stereotyped neuroanatomical locations. Examples of such areas are fusiform face area (FFA), extrastriate body area (EBA), and parahippocampal place area (PPA), which predominantly represent faces, bodies, and places, respectively[2].

A prominent yet unresolved question concerns strength of genetic and environmental influences on the organization of category-selective areas. Some studies have provided evidence in favor of an innate categorical organization. Face-versus-place selectivity can be detected in inferior temporal (IT) cortex in human infants by 5−6 months of age[3] and in macaque monkeys as young as 1 month of age[4]. However, monkeys raised without exposure to faces cannot develop normal "face patches" in IT cortex—suggesting that face experience is necessary for the formation of face domains[5]. Further evidence for the role of experience in the development of category-selective areas comes from other fMRI studies reporting that FFA and PPA are larger in human adults than children[6,7]. However, using auditory stimuli representing different categories, recent studies have demonstrated a similarity in the functional organization of category-selective ventral temporal cortex in congenitally blind subjects and sighted controls—arguing that the development of category-selective map in visual cortex does not rely on visual input and visual experience[8,9]. It has been proposed that the broad organization of ventral visual stream is driven by innate connectivity between regions that process semantic categories[10,11]. Finally, individuals with congenital prosopagnosia show lifelong difficulty in recognizing faces despite normal or almost normal exposure to faces, suggesting that the face recognition deficit in these individuals may have a genetic basis[12]. Genetic determination of face-processing system has also been suggested by studies reporting the heritability of face recognition behavior[13,14].

Given the conflicting results described above, a more direct approach for testing the role of genetic factors in category-selective cortex is needed. One important aspect of genetic influence is the heritability of individual differences, as addressed in classical twin studies. An fMRI study on identical (monozygotic, MZ) and fraternal (dizygotic, DZ) twins provides a unique opportunity for directly testing and disentangling the relative contributions of genes (nature) and environment (nurture) to the formation of category-selective areas. Using a small sample of twins (24 twin pairs), a previous study reported that the neural activity patterns in MZ twins were more similar than in DZ twins for the face and place stimuli[15]. Here we used a large sample of twins and a powerful analysis scheme (structural equation modeling[16]) to assess exactly the contribution of genetic factors to functional organization of category-selective areas, while carefully parsing out the contribution of genetic factors to structural similarity. Furthermore, we aimed to investigate the relationship between structural maps and fine-scale spatial maps of genetic influence in category-selective cortex.

## Results

We used fMRI data of 424 healthy young adults (212 twin pairs) from the Human Connectome Project (HCP) database (https://www.humanconnectome.org/study/hcp-young-adult). The twin subjects included 134 pairs of genetically confirmed MZ twins and 78 pairs of genetically confirmed, same-sex DZ twins. Each subject participated in a series of task fMRI scans. One of the tasks was a working memory task in which blocks of stimuli/pictures from four different visual categories (faces, bodies, places,

and tools) were presented. Category-selective maps were obtained by statistically comparing the activation for one category versus the average activation for the other three categories. For a given contrast, the maps were based on $z$-statistics. In this study, we considered face-, body-, and place-selective maps because these categories have specific representations in the human brain[2]. Maps from all subjects were projected onto a standard grayordinates space[17]. The grayordinates space contained 91,282 cortical and subcortical gray matter voxels/vertices.

A vertex-by-vertex intraclass correlation (ICC) analysis[18] was used to assess the similarity in category-selective maps between MZ twins and between DZ twins (Fig. 1a). For each vertex of the cortical surface and for a given category (e.g. face), the ICC value indicates resemblance (the degree of absolute agreement) between the $z$ values in the maps of twins. The ICC maps were visualized on a 2D flat patch of left and right hemispheres. Overall, the ICC values were greater in the MZ group than the DZ group, especially in occipital, posterior parietal, and posterior temporal cortices.

To quantify the effects, we ran a related analysis in which Pearson's correlation was computed between the whole-cortex activation patterns ($z$ maps) of one twin and the co-twin (Fig. 1b, leftmost panel). Then, the distribution of correlation coefficient values was plotted for all MZ and DZ twin pairs (Fig. 1b, right panels). In face-, body-, and place-selective maps, mean correlation was significantly higher in the MZ group than the DZ group. Mean correlation was also calculated separately for each cortical lobe (Fig. 1c). Consistent with the ICC maps, mean correlation was significantly higher in the MZ group than the DZ group in the occipital cortex.

Are face-, body-, and place-related activations genetically determined equally in all parts of visual cortex or are the genetic effects stronger in domain-specific areas? To address this question, we quantified the spatial overlap between the whole-cortex activation patterns in each twin pair, at different thresholds in the $z$ maps (Supplementary Fig. 1). The spatial overlap was higher in MZ twins than DZ twins, especially when only vertices/patches with higher category specificity were included in the maps. This result suggests that the genetic effects are stronger in domain-specific areas.

Next, we examined genetic influence on category-selective areas. For this, category-selective areas were first identified in the group-average maps of 787 subjects of the HCP database. Using objective thresholds for the maps (see Methods), we obtained distinct clusters of contiguous voxels that were analogous to known face, body, and place areas based on their anatomical locations (Fig. 2a and Supplementary Fig. 2). These areas included fusiform face area (FFA), occipital face area (OFA), medial face area (MFA), posterior and middle superior temporal sulcus face areas (pSTS and mSTS), amygdala face area (Amy), extrastriate body area (EBA), fusiform body area (FBA), parahippocampal place area (PPA), occipital place area (OPA), and medial place area (MPA)[1]. Place-selective activations were also localized within early visual areas V1, V2, and V3—though these activations may be related to the larger spatial extent of the place images compared to the other image types[19].

We then compared the correlation of activation patterns between MZ twin pairs, DZ twin pairs, and unrelated pairs in face, body, and place areas (Supplementary Fig. 3). The correlation was higher in MZ twins than DZ twins, and in DZ twins than unrelated pairs, suggesting a strong genetic influence in category-selective areas. This result was also consistent with the graded genetic similarity of MZ and DZ twins (MZ twins share all their genes, while DZ twins share on average half of their genes).

For genetic analysis, we employed path analysis, which is a common statistical technique used in twin studies[16]. In this

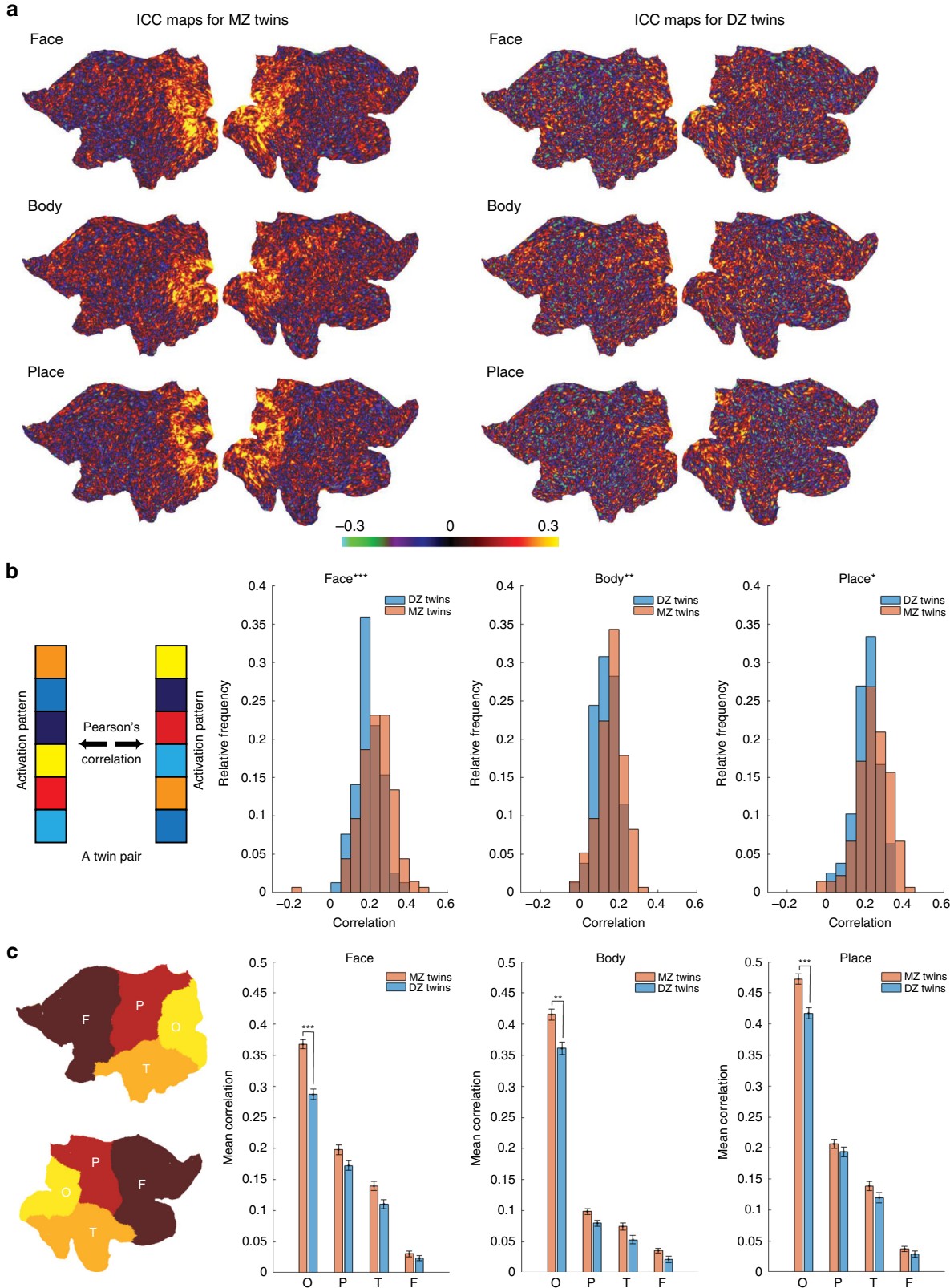

analysis, a genetic structural equation model (Supplementary Fig. 4) was fitted to the twin data in order to linearly decompose the observed variance of the phenotype of interest (here, category-selective activations) into genetic (A) and environmental (E) components.

Based on the AE model, the "genetic voxels" (voxels having a significant genetic influence) were determined (Supplementary Fig. 5; see also Methods). Figure 2b shows the maps of genetic voxels in face, body, and place areas after FDR correction. These maps, which were cross-validated (Supplementary Fig. 6),

**Fig. 1 Correlation between category-selective activations in MZ and DZ twins. a** For each category, ICC maps in MZ and DZ twins are displayed on a 2D flat patch of left and right hemispheres (left and right maps in each subpanel). **b** Left panel: A schematic diagram depicting the procedure for computing Pearson's $r$ correlation. In each twin pair, the activation patterns ($z$ maps) of one twin and the co-twin were correlated. In the activation vector, data from two hemispheres were concatenated. Right panels: Distribution of correlation coefficient values for all MZ and DZ twin pairs in face-, body-, and place-selective maps. **c** Mean correlation was calculated separately for each cortical lobe (O occipital, P parietal, T temporal, F frontal). Lobar parcellation was based on PALS-B12 atlas of human cerebral cortex[55] (http://brainvis.wustl.edu/wiki/index.php/Caret:Atlases). Error bars represent one standard error of the mean, here and in the other figures. *$p < 0.05$, **$p < 0.005$, ***$p < 0.0005$; Bonferroni-corrected pooled $t$ test for three comparisons in panel (**b**) and 12 comparisons in panel (**c**).

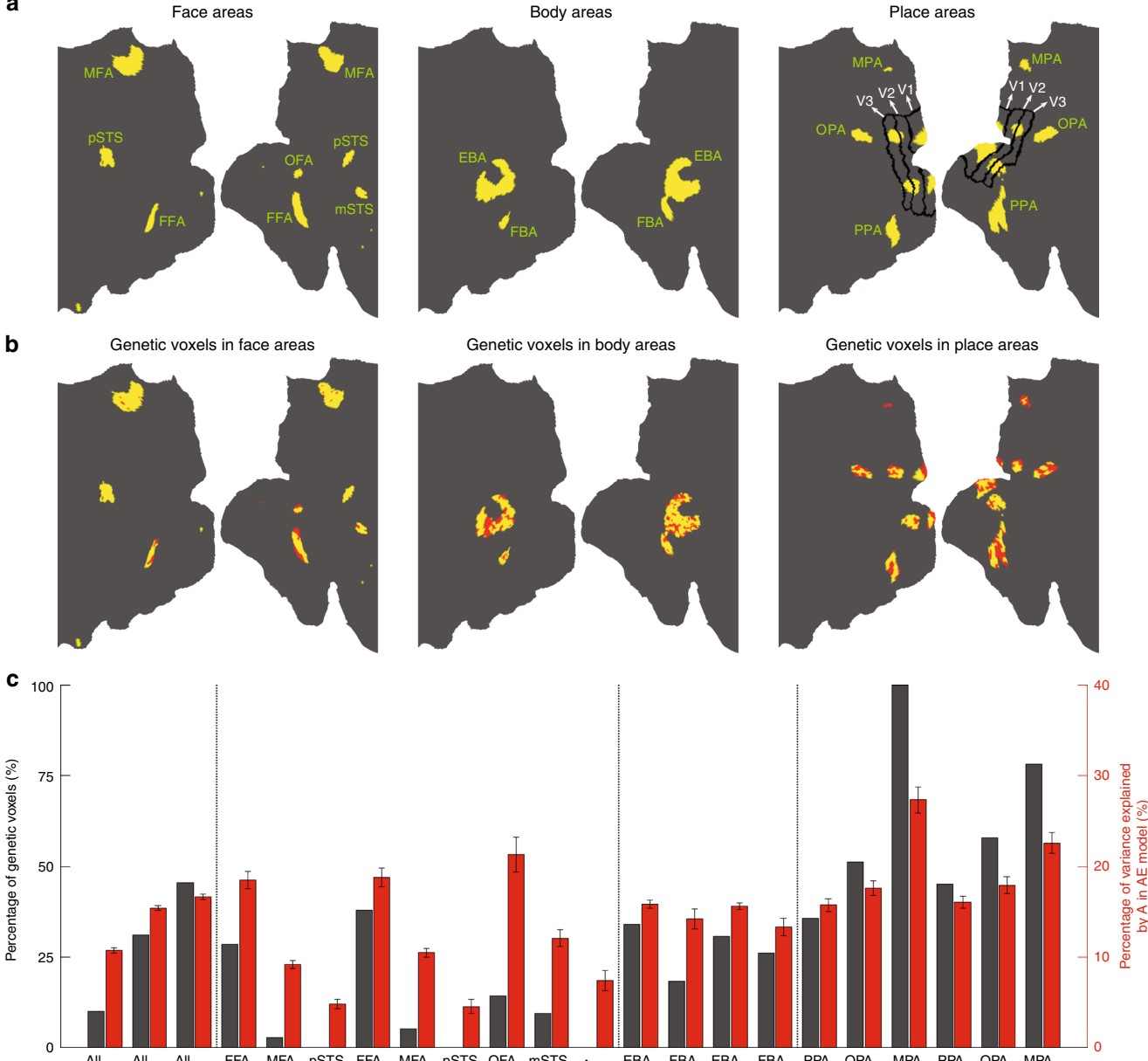

**Fig. 2 Genetic analysis in category-selective areas. a** Face, body, and place areas are displayed on a 2D flat patch of left and right hemispheres (left and right maps in each subpanel; see text for the full name of areas). The face-selective area in the posterior cingulate cortex[56] was named MFA (medial face area) here. The areal borders of V1/V2/V3 were estimated based on a probabilistic map of retinotopic areas in a group of 12 subjects[57]. Place-related activations in V1/V2/V3, which avoided the V1/V2 border, were localized in/near regions representing the mid-peripheral visual field. For each category, there were 913 category-selective voxels. **b** Voxels with a significant genetic effect (FDR-adjusted $p < 0.05$ based on 913 comparisons) are marked red in the face, body, and place maps. For these voxels, the average A value was ~25%. **c** Region-of-interest analysis demonstrating descriptive statistics for the genetic effect in category-selective networks (all areas together) and in individual category-selective areas. For this analysis, EBA and FBA in the right hemisphere, which were conjoined in the map, were separated at their junction. The effect of A varied significantly ($p < 0.0005$; one-way ANOVA) across networks, across face areas, and across place areas.

revealed interesting features. First, a considerable number of genetic voxels was observed in FFA, OFA, all body areas, and all place areas, as quantified in Fig. 2c. Second, the genetic influence was not homogeneous within the category-selective areas, suggesting that the genetic effects could vary at a fine spatial scale—with nearby subregions having different genetic effects. Third, the genetic voxels were relatively clustered in FFA, OFA, and place areas (e.g. in posterior + medial part of FFA and central part of PPA), while they appeared to be scattered throughout the body areas.

Consistent with a low percentage of genetic voxels in the whole network of face areas (all face voxels together), the effect of A (the percentage of variance explained by A in AE model) was significantly lower in the face network than the body and place networks (Fig. 2c). The effect of A varied significantly across face areas and across place areas—though the pattern of variation across these areas was strikingly similar in the two hemispheres (Fig. 2c and Supplementary Fig. 7). Among face areas, the genetic effect was strong only in FFA and OFA. Among place areas, MPA showed the highest genetic effect.

Using structural data from all twin subjects, we explored whether the pattern of genetic voxels within category-selective areas has any systematic relationship with the pattern of cortical folding/curvature, thickness, and myelination in those areas. For this, we compared the maps of genetic effect with the maps of structural properties which were obtained for an average ($n = 424$) cortical surface (Fig. 3a). In face and place areas, genetic voxels were preferentially concentrated in regions with higher cortical curvature (gyral crowns in face areas and sulcal fundi in place areas). Genetic face voxels showed a significantly higher cortical thickness and myelination, compared to nongenetic face voxels. Conversely, genetic place voxels showed a significantly lower cortical thickness and myelination, compared to nongenetic place voxels. Such structural biases were not detected in body areas.

The observed genetic effects in category-selective voxels were primarily due to more similarity of functional activation patterns in MZ twins than DZ twins. However, zygosity could have different effects on anatomical similarity as well. To address this concern, we looked at the heritability of cortical curvature, thickness, and myelination in category-selective areas. These structural measures were more similar in MZ twins than DZ twins, as revealed by the ICC maps (Fig. 3b); however, the patterns of anatomical similarity were qualitatively different from the patterns of functional similarity (see Fig. 1a). Genetic analysis in category-selective areas provided a quantitative confirmation that voxels showing the heritability of cortical structure had little or no overlap with voxels showing the heritability of category-selective activation (Fig. 3c). Thus, the genetic influence on category selectivity could not be attributed to the heritability of macroscopic structural features per se.

How much of the greater similarity in spatial patterns of functional response for MZ than DZ twins results simply from the overall greater similarity in the shape of the brain in MZ than DZ twin pairs? To address this question, we measured the similarity of the transformations required to register the two co-twin brains to the standard brain used in our study. Using three different metrics, we found that the registration-induced distortion maps were more similar between two MZ than between two DZ twins (Supplementary Fig. 8a). However, an additional analysis showed that this similarity in the brain shape of MZ twins was a ubiquitous phenomenon that was present in both "genetic" and "nongenetic" category-selective cortex (Supplementary Fig. 8b). Thus, the similarity of functional activation patterns in MZ twins was above and beyond the mere similarity in the brain shapes.

## Discussion

The comparison of MZ and DZ twins in our study revealed the heritability of category-specific representations in visual cortex. Face-specific activations appeared to be only modestly heritable. FFA and OFA showed strong genetic influence, as they were located in regions with high ICC values. However, genetic influence was weak in other face-selective areas (e.g. pSTS and MFA). Genetic influences have also been reported for other aspects/measures of cortical organization including cortical surface area[20], cortical thickness[21], cortical regionalization[22], structural and functional connectivity architecture[23–25], functional activations[26], and brain rhythms[27]. Using a small sample of twins, one study reported that the neural activity patterns in MZ twins were more similar than in DZ twins for the face and place stimuli[15]. Here using a large sample of twins, precise anatomical localization and genetic modeling, we discovered that the genetic influence was not homogeneous across the category-selective areas; only parts of these areas showed significant genetic effect. Strikingly, genetic subregions within face and place areas contained specific structural properties (i.e., the genetic influence had a link with the pattern of cortical curvature, cortical thickness, and cortical myelination in these areas).

Face and place recognition have been ecologically important cognitive tasks in primates throughout evolution. Converging evidence from human and macaque fMRI studies suggests that these two primate species have a homologous cortical architecture for the processing of faces and places[28–31]. Thus, it is plausible that the organization of face- and place-selective areas is at least partially dictated by genetics. The link between genetics and cortical folding could then be explained by tension-based theory of cortical morphogenesis[32]. According to this theory, cortical connections between functionally related areas are formed during the prenatal stage of cortical development (see also ref. [10]). Mechanical tensions along axons of these connections would force cortically adjacent regions to get closer to each other, which would subsequently induce the formation of gyral folds. Axonal tensions may also pull remotely interconnected regions towards one another, thus contributing to the formation of sulcal folds. These mechanisms could possibly explain why genetic face and place voxels are more prevalently found in regions with higher folding/curvature.

Previous fMRI studies have shown that the representation of visual field eccentricity (a low-level visual feature) extends into higher-tier category-selective cortex[33]. For instance, the face-selective FFA and the word-selective visual word form area[34] contain a foveal representation, whereas the place-selective PPA contains a peripheral representation. The heritability of category-specific representations could be, to some extent, related to the heritability of lower-level eccentricity representations. In the case of word area, evidence suggests that word-selective activations are heritable[35], and that visual experience is not needed for the formation of this area[36,37]. Reading is not an evolutionarily old skill. Thus, it is unlikely that the word selectivity itself has a genetic basis. However, a more primitive form of representation in this area (namely, the foveal representation) could have emerged through innate mechanisms. In the case of face, body, and place areas, it remains an open question how much of the heritability in the activation patterns is related to the heritability of eccentricity representations.

An increased cortical thickness in genetic face voxels suggests that these voxels may contain a higher number of neurons, perhaps for a "sparse coding" of faces. As shown previously, tissue development and microstructural proliferation in face-selective (but not place-selective) cortex is coupled with increases in face selectivity and improvements in face recognition[38]. On the other hand, myelin content has been shown to be inversely correlated

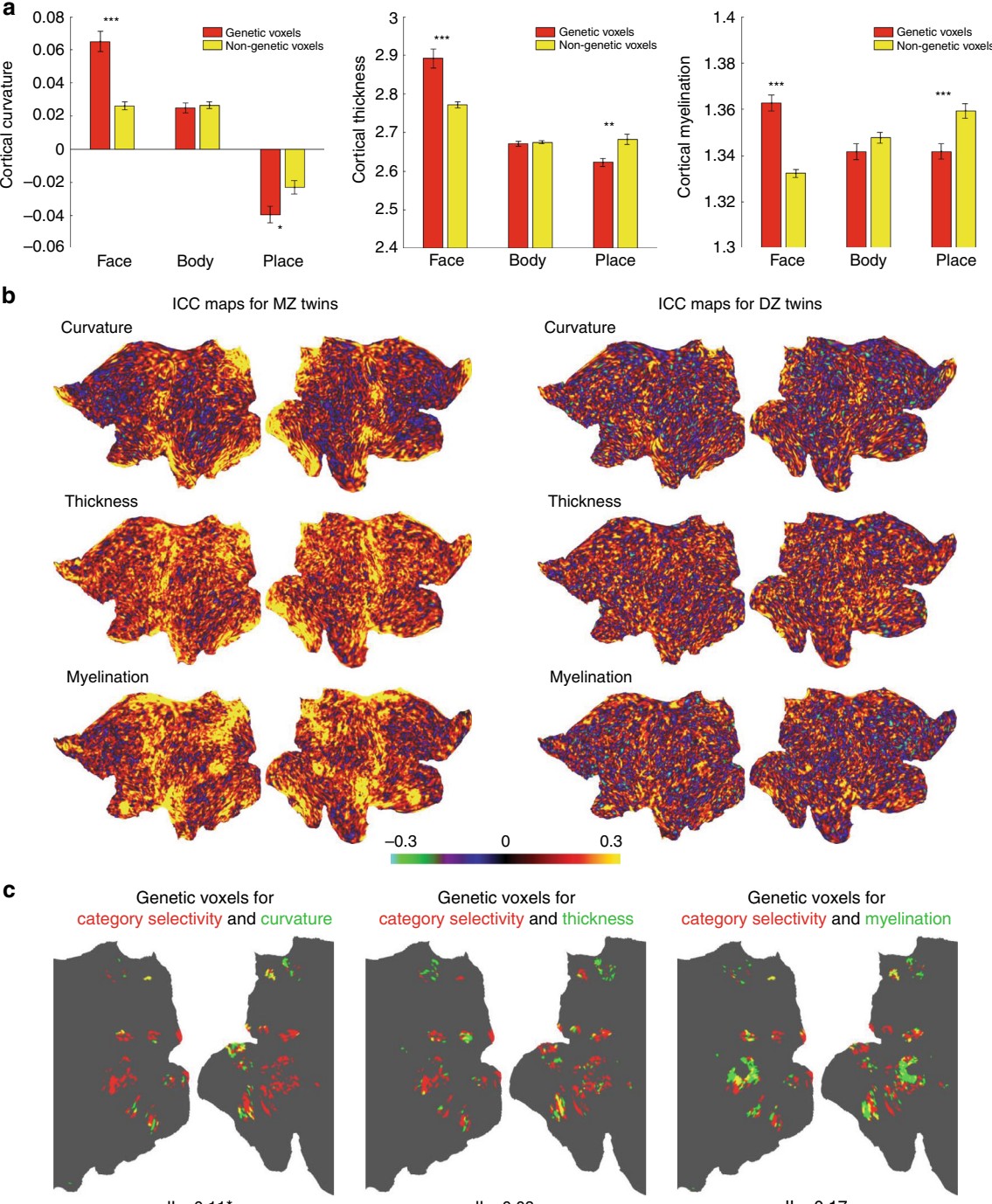

**Fig. 3 Relationship between genetic effect in category-selective areas and structural/morphological properties of cortical gray matter. a** In face, body, and place areas, mean cortical curvature, mean cortical thickness, and mean cortical myelination were computed across genetic and nongenetic voxels. Place-selective voxels in V1/V2/V3 were excluded in this analysis. Regions with positive curvature corresponded to cortical gyri, and regions with negative curvature corresponded to cortical sulci. *$p < 0.05$, **$p < 0.005$, ***$p < 0.0005$; Bonferroni-corrected pooled $t$ test for three comparisons between genetic and nongenetic voxels in each plot. **b** ICC maps for curvature, thickness, and myelination in MZ and DZ twins, displayed on a 2D flat patch of left and right hemispheres. **c** Using the same procedure for estimating the heritability of category selectivity (see Methods), the heritability of curvature, thickness, and myelination was estimated in each category-selective voxel. The maps show voxels with genetic influence on category selectivity (red), cortical structure (green), or both (yellow). The degree of overlap between the binary genetic influence maps was quantified using the Jaccard Index (JI). JI was low in all cases. In the case of curvature, it was slightly and significantly above the expected JI under independence ($p < 0.05$; bootstrap test). In the case of thickness and myelination, the difference was not significant ($p > 0.05$; bootstrap test).

with intracortical circuit complexity[39]. Thus, a decreased cortical myelination in genetic place voxels suggests that these voxels may have more dynamic intracortical circuits, perhaps for a "distributed coding" of places.

Further studies are needed to test whether the density of neurons and their connections is indeed different in genetic and nongenetic subregions of category-selective areas. New advances in diffusion MRI technique (e.g. "neurite imaging"[40]) makes it possible to investigate fiber density within gray matter at a high resolution. Furthermore, recent databases such as the whole-brain gene expression maps provide a unique platform for screening genes and relating them to the cortical organization[41]. Exploring these databases may shed light on how genes play a causal role in the formation and development of category-selective areas.

## Methods

**Subjects**. In this study, we used the "HCP1200" dataset (March 2017 data release) of healthy adults aged 22−35 (https://www.humanconnectome.org/study/hcp-young-adult/document/1200-subjects-data-release). The dataset included 424 twin subjects (252 females, 172 males). Of 212 twin pairs, 134 pairs were genetically confirmed MZ twins and 78 pairs were genetically confirmed, same-sex DZ twins. Subjects were recruited from Washington University (St. Louis, MO) and the surrounding area. The HCP data were acquired using protocols approved by the Washington University institutional review board, and written informed consent was obtained from all subjects.

**Data acquisition**. The HCP MRI data acquisition has previously been described in detail[42–44]. Images were acquired using a customized 3T Siemens "Connectom" Skyra scanner having a 100 mT/m SC72 gradient insert and a standard Siemens 32-channel RF-receive head coil. At least one 3D T1w MPRAGE image and one 3D T2w SPACE image were acquired at 0.7 mm isotropic resolution. Whole-brain resting-state fMRI and task fMRI data were acquired using multiband EPI sequence with parameters of TR = 720 ms, 2 mm isotropic voxels, and multiband acceleration factor of 8. Spin echo field maps were acquired during both structural and fMRI scanning sessions to enable accurate cross-modal registration of structural and functional images in each subject.

**Task paradigm**. Functional data in this study were based on the HCP working memory task[19]. It was a version of the N-back task to assess working memory and cognitive control. By presenting blocks of trials that consisted of pictures of faces, places, tools, and body parts, this task could also be used as a "functional localizer" to obtain category-specific representations[45].

Subjects performed two runs of the working memory task. Each run contained eight task blocks (25 s each) and four fixation blocks (15 s each). The four different stimulus types (faces, places, tools, and body parts) were presented in separate task blocks. Each task block contained ten trials. On each trial, the stimulus was presented for 2 s, followed by a 500 ms inter-trial interval. Within each run, four blocks used a 2-back working memory task (respond "target" whenever the current stimulus was the same as the one 2-back) and the other four blocks used a 0-back working memory task (respond "target" whenever the current stimulus was the same as the target stimulus presented at the start of the block). A 2.5 s cue indicated the task type (and target for 0-back) at the start of the block. In each block, there were two targets and 2–3 nontarget stimuli (repeated items in the wrong n-back position, either 1-back or 3-back).

**Data analysis software**. Data were preprocessed and analyzed using the publicly released HCP pipelines[42]. The software packages used for analysis included Connectome Workbench commandline tools, FreeSurfer, and FSL[46,47]. Connectome Workbench "wb_view" GUI (http://www.humanconnectome.org/software/connectome-workbench.html) was used for visualization of maps and creating ROIs.

**Analysis of structural data**. Structural images (T1w and T2w) were used for extracting subcortical structures and reconstructing cortical surfaces in each subject. Volume data were transformed into MNI space using a nonlinear volume-based registration. For accurate cross-subject registration of cortical surfaces, a multimodal surface matching (MSM) algorithm[48,49] was used. The MSM algorithm had two versions: "MSMSulc" (nonrigid surface alignment based on folding patterns) and "MSMAll" (optimized alignment of cortical areas using sulcal depth maps plus features from other modalities including myelin maps, resting-state network maps, and visuotopic connectivity maps). Data in our work were based on MSMSulc registration. We obtained similar results when data were based on MSMAll registration. After surface and volume registration, cortical vertices were combined with subcortical gray matter voxels to form the standard "CIFTI grayordinates" space (91,282 vertices/voxels with ~2 mm cortical vertex spacing and 2 mm isotropic subcortical voxels).

For each point/vertex of the cortical surface at the boundary of gray matter and white matter, the measurement of cortical curvature was based on mean curvature: the average of principal curvatures derived from the inverse of the radius of the osculating circles at that point. Cortical thickness was measured as the distance between gray matter and white matter. Because gyral crowns tend to be thicker than sulcal fundi, the cortical thickness values were corrected for folding-related biases by regressing out the mean curvature from thickness data[50]. Myelin maps were computed using the ratio of T1w/T2w image intensities, and they were corrected for the residual bias field present in the image[39,50].

**Analysis of fMRI data**. Functional images were minimally preprocessed in the HCP pipeline[42]. The preprocessing included correction for spatial distortions due to gradient nonlinearity and b0 field inhomogeneity, fieldmap-based unwarping of EPI images, motion correction, brain-boundary-based registration of EPI to structural T1w scan, nonlinear registration into MNI space, and grand-mean intensity normalization. Data from the cortical gray matter ribbon were projected onto the surface and then onto the standard grayordinates space. Subcortical data were also projected to a set of subcortical gray matter structures in the grayordinates space. Data were minimally smoothed by a 2 mm FWHM Gaussian kernel in the grayordinates space (smoothing was constrained to cortical surface and subcortical gray matter parcels). Data were cleaned up for artifacts and structured noise using ICA + FIX.

The preprocessed functional time series were entered into a general linear model (GLM) to estimate functional activities in each vertex/voxel in each run[19]. For the working memory task, eight regressors/predictors were used in the GLM design—one for each type of stimulus in each of the N-back conditions. Each regressor covered the period from the onset of the cue to the offset of the final trial (27.5 s). All regressors were convolved with a canonical hemodynamic response function and its temporal derivatives. The time series were temporally filtered with a Gaussian-weighted linear highpass filter with a cutoff of 200 s, to remove low-frequency drifts/fluctuations presumably unrelated to the task design. The time series were also prewhitened to remove temporal autocorrelations in the fMRI data. Linear contrasts were computed to estimate effects of interest: each stimulus type versus all others, collapsing across memory load. Fixed-effects analyses were conducted to estimate the average effects across runs within each subject, then mixed-effects analyses treating subjects as random effects were conducted to obtain group-average maps.

The category-selective voxels were defined as the top 1% of voxels (913 out of 91,282 voxels) which had the highest $z$ values in a given contrast (e.g. faces versus all other categories). The 99th percentile corresponded to the cutoff-point $z$ values of 12.38, 16.89, and 27.35 in group-average face, body, and place maps, respectively. The category-selective voxels were almost identical using MSMSulc versus MSMAll registrations (Supplementary Fig. 9). One minor difference between MSMSulc and MSMAll maps was that, in the right hemisphere, the "middle STS face area" was better localized in the MSMSulc map while the "anterior STS face areas" were better localized in the MSMAll map.

**Genetic modeling**. For genetic modeling of twin data, a univariate structural equation model was constructed in the statistical package OpenMx[51,52]. The components of the model included variance caused by additive genetic factor (A), dominance genetic factor (D), and environmental factors (see Supplementary Fig. 2). The environmental effects were divided into those that were shared in common by members of a twin pair (C) and those that were unique to each twin (E, including measurement error).

To determine which model (ADE or ACE/AE) was the most appropriate one for our data, the ICC correlation data in MZ and DZ groups (see Fig. 1a) were used to estimate the broad sense heritability [$h^2 = 2*(r_{MZ} − r_{DZ})$] and common environmental influence [$c^2 = (2*r_{DZ}) − r_{MZ}$] in each category-selective vertex/voxel[53], then the distributions of $h^2$ and $c^2$ were plotted (see Supplementary Fig. 3). In face, body, and place areas, the $h^2$ values were generally greater than zero, consistent with the presence of genetic influences (heritability). In some voxels, the $c^2$ values were negative (i.e., the DZ correlations were less than half the MZ correlations), suggesting genetic dominance effect in those voxels. However, many voxels (~50% of voxels) did not show genetic dominance effect. Thus, there was not enough support for the ADE model.

To compare ACE and AE models, we calculated a model-fit metric (the Akaike Information Criterion, AIC) for ACE and AE models in all category-selective voxels. In almost all voxels, the AIC of AE model was less than the AIC of ACE model (average ΔAIC = 19.42). Since the ACE model showed significantly worse fit to the data compared to the AE model (ΔAIC > 10[54]), we selected the more parsimonious AE model—with A including all genetic factors, and E including all environmental factors.

In each category-selective voxel, the relative contribution of each latent factor (A and E), expressed by the regression path coefficients $a$ and $e$, was estimated using the maximum likelihood criterion. The variance related to each factor was defined as the square of the path coefficients, $a^2$ and $e^2$. Thus, the percentage of variance explained by A in AE model was defined as $a^2/(a^2 + e^2) \times 100$. For each voxel, the effect of genetics was considered significant if dropping A from the

model (i.e., using the submodel E) resulted in a significant decrease in the goodness-of-fit $\chi^2$ statistic.

**Reporting summary**. Further information on research design is available in the Nature Research Reporting Summary linked to this article.

## Data availability

All data used in this manuscript are part of publicly available and anonymized HCP database (https://www.humanconnectome.org).

## Code availability

All analysis codes are available for sharing upon request.

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

## Acknowledgements

We thank Rouhollah Abdollahi for help with data analysis, and David C. Van Essen and Moataz Assem for helpful comments on the manuscript. This research was supported by Cognitive Science and Technology Council of Iran (CSTC), Iranian Institute for Research in Fundamental Sciences (IPM), University of Cambridge—MRC Cognition and Brain Sciences Unit, Cambridge Trust scholarship, and Preston Robb fellowship.

## Author contributions

R.R. conceived the ideas and designed the analyses; N.A. performed the analyses and prepared the figures; R.R. wrote the manuscript; J.D. critically revised the manuscript. All authors approved the final version of the manuscript.

## Competing interests

The authors declare no competing interests.
