## [Peer Review File · Nature Communications]

Reviewers' Comments:

Reviewer #1:

Remarks to the Author:

In their manuscript "Genetic influence is linked to cortical morphology in category-selective areas of visual cortex", Abbasi et al. uses a large data set from 1200 participants, including mono- (MZ) and dizygotic (DZ) twins, from the human connectome project to determine the similarity of functional maps for three main categories, faces, bodies, and places, compared between the two twin groups, and their relationship to cortical folding.

This is a highly interesting and clear-cut study that deserves to be published in Nature Communications. The overall study design is very elegant, and the combination of functional with curvature data intriguing, and impactful conclusions can be drawn from the results.

I have a few concerns I hope the authors can address:

My main concern relates to the conclusion of the paper based on the different analyses. In Fig. 1 we see specific increases in correlations for the MZ over the DZ group in occipital cortex for faces, bodies, and places. While correlations for bodies and places are, overall, higher than for faces, the MZ-DZ differential effect, if anything, appears to be stronger for faces than the other two categories. The author then continue to identify "genetic" voxels in a second analysis and find that there are fewer of these voxels in face than in body and place areas. If I am not mistaken, then this difference between domains could be reconciled with the results from the first analysis, by assuming that the genetic influence in face areas spreads out more and reaches significance in fewer voxels, while it is more focal in the other domains. But the main conclusion of the paper, that face areas are less genetically determined than body and place areas, seems to rely on the summary statistics of genetic voxels alone. I am not sure that this conclusion is warranted.

A second concern regards the fact that all three domains appear to be genetically determined to some extent. Why would that be the case? Is it that functional domains that have been easy to guess by experimenters as possibly involving functional specialization and lead to very focal activation happen to also be genetically more determined than other functions? Or is it that, maybe, all parts of the brain are equally genetically determined, just that we have an easier time with the spatial resolution of the functional maps we can obtain and the inherent errors of cross-registration etc. revealing these genetic contributions in these domain specific areas? It would be great if the authors could run a simulation of the latter possibility and generate an argument that this would not provide an explanation for their results. (This is more a request than a requirement.) I think this would be important for interpretation. It would be great if there were other functional specializations that do not show the effect. For example, can the authors analyze object-selective area LOC and show that it is not genetically determined?

Relatedly, I am wondering what about other, higher-level functional specializations and whether they correlate. Given that subjects in this study were engaged in a number of different tasks, higher-order social cognition or speech processing could serve as a comparison. Alternatively areas thought to support general intelligence might serve as a point of comparison. Again, the point is to show whether there are discrete functions that might not show a genetic advantage.

Reviewer #2:

Remarks to the Author:

The authors report the results of a large scale reanalysis of data available through HCP to test whether there is greater similarity in activity patterns and structural brain morphology between monozygotic twin pairs compared to dizygotic twin pairs. The principal finding is that there is a significant contribution of genetics to similarity in category-specific areas of the ventral stream.

This is an important contribution on an important topic. The issues highlighted below are primarily focused on the need for the theoretical contribution of the paper to be better situated in the setting of the prior literature. I also found it surprising that the authors did not use an 'unrelated' participant pair baseline: i.e., it seems that the study design begs for monozygotic, dizygotic and unrelated pairs.

1. The authors only point out midway through the paper that a prior study (Polk et al) found similar findings in terms of functional activity; that prior study was within far fewer subjects, but its not clear if the authors expansion of experimental power is more than an incremental advance. The key issue that the authors need to specify what is theoretically novel about their contribution—my sense is that combination of analysis of functional activity and gyral folding and myelination frames a potentially important theoretical contribution that is hinted at but not fully developed.

2. Why did the authors not carry out the analyses for unrelated pairs (e.g., reshuffles on the twin data to make genetically and experientially unrelated pairs)? If there is an effect of genetics then it should be greater for dizygotic than unrelated pairs, right? This seems like an important validation and extension that is not tested even though the authors have the data.

3. Large swaths of the prior literature are not acknowledged; this has the effect of weakening the contribution of the study rather than making it seem more novel. For instance, the authors reference a single study with congenitally blind subjects, but there is a lot more work in that space that has already made the argument that the broad organization by semantic category in the ventral stream is driven by innate constraints, and that those innate constraints pertain to the connectivity of the ventral stream. This argument about the role of innate connectivity constraints on shaping the organization of ventral stream was initially made by Caramazza and coauthors (see refs below) and subsequently picked up by Kanwisher and colleagues and others. Those arguments may also tie into the discussion in the GD about axonal tension and cortical folding.

4. The framing of the paper rests on a false dichotomy: the dichotomy that is set up in the introduction is between 'genes' and 'experience'. The dichotomy is false because there are not theories of innate constraints that do not require experience (i.e.. development). Even something as prosaic as 'growing a hand', which is clearly genetic, requires an environment (ie experience). A little closer to home—if you assume human language results from a genetic endowment, that theory is not embarrassed by observations that children (tragically) raised without linguistic input do not develop normal language abilities. There is no zero sum between genetics and experience; the question is whether there is a contribution of genetics, quite independent of whether there is a contribution of experience. So the results from Acaro et al, for instance, do not show that face specificity is not genetically constrained—just that experience is necessary (but not genetic story denies you need experience). More subtle engagement with these asymmetries would greatly strengthen the paper. An excellent example of this playing out in fact is the debate about 'congenital' prosopagnosia, and whether it points to genetic contributions to face specificity (see for instance, recent exhaustive review with commentaries, by Geskin and Behrmann, 2018).

5. The GD missed many opportunities to reinforce the theoretical contribution of the paper—the discussion of sparse versus distributed coding of faces and places, based on cortical thickness, seems incredibly speculative and ungrounded. I would encourage the authors to focus their discussion on what their paper is directly measuring and about—which is whether there are genetic contributions to category-specificity.

For instance, Buchel et al and then Striem-Amit et al have shown quite clearly that there is a visual word form area in the congenitally blind. Reading is not an evolutionarily old skill. So it cannot be that 'reading' is innate. And yet, visual experience is not needed for emergence of a reading area (in the

blind it responds to braille). So clearly there is some non-visual constraint here—and probably (what else could it be?) it is the connectivity that is innate.

Would the authors predict greater similarity in the visual word form area in monozygotic than dizygotic twins? I think they would based on the data from the blind—so what would this mean for their argument?

References the authors may find useful (this is in no way exhaustive—I would encourage the authors to thoroughly look into the precedent of their claim for a genetic contribution to category-specificity in the ventral stream).

Buchel, C., Price, C., and Friston, K. (1998). A multimodal language region in the ventral visual pathway. *Nature* 394, 274–277.

Geskin and Behrmann. 2018. Congenital prosopagnosia without object agnosia? A literature review. *Cognitive Neuropsychology*, 35, 4-54.

Mahon, B.Z. & Caramazza, A. (2011). What drives the organization of object knowledge in the brain? *Trends in Cognitive Sciences*, 15, 97-103.

Mahon, B.Z., Anzellotti, S., Schwarzbach, J., Zampini, M., & Caramazza, A. (2009). Category-specific organization in the human brain does not require visual experience. *Neuron*, 63, 397-405.

X Wang, C He, MV Peelen, S Zhong, G Gong, A Caramazza, Y Bi. Domain selectivity in the parahippocampal gyrus is predicted by the same structural connectivity patterns in blind and sighted individuals. *Journal of Neuroscience* 37 (18), 4705-4716

Striem-Amit E, Cohen L, Dehaene S, Amedi A. Reading with sounds: sensory substitution selectively activates the visual word form area in the blind. *Neuron* 2012;76 (3) :640-52.

Saygin, Z.M., Osher D.E., Norton E.S., Youssoufian D.A., Beach S.D., Feather J., Gaab, N., Gabrieli, J.D.E., & Kanwisher, N. (2016). Connectivity precedes function in the development of the visual word form area. *Nature Neuroscience*, 19(9):1250-5.

Reviewer #3:

Remarks to the Author:

Abassi et al report an important set of findings on the heritability of category-selective responses in the cortex, using the Human Connectome Project database. They find that the spatial patterns of selectivity for faces, places, and bodies are more correlated between pairs of monozygotic (MZ) than dizygotic (DZ) twins, i.e., these spatial patterns of category selectivity are heritable. They further show that the most heritable voxels (“genetic voxels”) in the cortex within category-selective regions were located in voxels with higher curvature, and that that cortex was thicker and more myelinated in genetic voxels of face areas, while it was thinner and less myelinated in genetic voxels of place areas. The paper is clearly written, and the results will be of interest to a wide neuroscience audience. I had only one significant concern, and a few other questions and suggestions. Assuming the authors can address the significant concern (#1 below), I think the paper deserves to be published in *Nature Communications*.

1. How much of the greater similarity in spatial patterns of functional response for MZ than DZ twins results simply from the overall greater similarity in the shape of the MZ than DZ twin pairs? If DZ brains are more different in overall shape than MZ twins, then the cortical alignment is likely to be not as good for DZ as MZ twins, and as a result the functional correlations will likely be lower. The authors do show that some anatomical features of brains are heritable (like curvature and myelination; Figure 3). But the question here is whether there are heritability effects of the patterns of functional response over and above any anatomical differences in the mere shapes of the brains. One way to answer to this question would be to measure the similarity of the transformations required to register the two co-twin brains to the standard brain used the study. If these transformations were more similar between two MZ than between two DZ twins, then the worry would remain that the apparently greater similarity in spatial patterns of functional response simply results from more consistent alignment to the standard brain for MZ than DZ twins. If the authors think this is not a reason for concern clearly explain why to readers. (On a related note the authors should fix the colormap axis to the same upper threshold of 0.4 in both Figure 1 and 3 to aid the qualitative comparison between these plots).

2. What exactly does heritability mean?

Although I did not see an misinterpretation of the meaning of heritability, I think the paper would benefit from the addition of a sentence or two clarifying what exactly heritability does and does not mean. As others have written in detail (see for example, Block, *Cognition*. 1995 Aug;56(2):99-128.), what we really want to know about the brain is genetic determination: that is, which properties of the brain are determined by genes, versus which result from experience. But that is not what heritability measures. Heritability measures how much of the variation between individuals is accounted for by genes, and how much by experience. Heritability can diverge importantly from genetic determination. For example, as Block notes in the reference above, having five fingers is clearly genetically determined. However, the heritability of finger number is almost zero, because most of the variation found in finger number in the population is most likely due to experience (in this case accidents), not genes. As another way in which heritability of a given characteristic does not at all mean that that that characteristic is genetically determined, note that the genetic influence found in this study could exert its effects in all kinds of indirect ways, including via experience. As just one example, perhaps what is heritable is the subject's eye fixation behavior on faces (and places and bodies). If so, the precise retinotopic location of different features during experience would be heritable, and perhaps with it the precise location of that selectivity in the ventral visual pathway. In this case the heritability of the cortical patterns reported here could reflect primarily an experiential effect in disguise, operating via a genetic effect on eye fixation behavior. That is just one of many possible examples of how a heritability effect can be mediated by experience. I would like to see the authors mention some of these subtleties in the papers that others will not take these results as indicating genetic determination of the category selective regions themselves.

3. The main question, analysis, and results from the paper are very similar to those from the Polk et al study from 2007, so one might worry about how novel the paper is. I would actually argue that the N is so much larger here, and the result so worth replicating, that the paper is nonetheless worthy of publication in *Nature Communications*. If our field is to get its act together we need to place more weight on solid results and replications than on novelty. Further, the relationship to cortical curvature is novel, as is the analysis of cortical thickness and tissue myelination within face and place areas.

Other Minor Points

There may not be room in this short format, and I do not at all insist, but it would also be nice to see the authors mention some other things:

i) brief mention of the heritability of face recognition behavior, which several studies have reported:

<https://www.ncbi.nlm.nih.gov/pubmed/20176944>
<https://www.ncbi.nlm.nih.gov/pubmed/20176944>
<https://www.ncbi.nlm.nih.gov/pubmed/20060296>

ii) How might the results of the current study be reconciled with the Arcaro et al findings? I think they can be, easily, but it might help to spell this out.

Small point: If the medial face area (MFA) has been reported before, please include a citation.

Nancy Kanwisher

Response Letter

Reviewer 1

In their manuscript “Genetic influence is linked to cortical morphology in category-selective areas of visual cortex”, Abbasi et al. uses a large dataset from 1200 participants, including mono- (MZ) and dizygotic (DZ) twins, from the human connectome project to determine the similarity of functional maps for three main categories, faces, bodies, and places, compared between the two twin groups, and their relationship to cortical folding. This is a highly interesting and clear-cut study that deserves to be published in Nature Communications. The overall study design is very elegant, and the combination of functional with curvature data intriguing, and impactful conclusions can be drawn from the results.

We thank the reviewer for a positive assessment of our manuscript.

I have a few concerns I hope the authors can address:

My main concern relates to the conclusion of the paper based on the different analyses. In Fig. 1 we see specific increases in correlations for the MZ over the DZ group in occipital cortex for faces, bodies, and places. While correlations for bodies and places are, overall, higher than for faces, the MZ-DZ differential effect, if anything, appears to be stronger for faces than the other two categories. The authors then continue to identify “genetic” voxels in a second analysis and find that there are fewer of these voxels in face than in body and place areas. If I am not mistaken, then this difference between domains could be reconciled with the results from the first analysis, by assuming that the genetic influence in face areas spreads out more and reaches significance in fewer voxels, while it is more focal in the other domains. But the main conclusion of the paper, that face areas are less genetically determined than body and place areas, seems to rely on the summary statistics of genetic voxels alone. I am not sure that this conclusion is warranted.

The ICC maps in Figure 1 qualitatively show that the functional activations are more similar in the MZ group than the DZ group, in the occipital cortex. As mentioned by the reviewer, this similarity exists for all three categories – though the magnitude and spatial extent of correlation appears to be slightly weaker for body than face and place (see also Figure 1 in the response letter). In the genetic analysis of category-selective areas in Figure 2, FFA and OFA show strong genetic influence. FFA and OFA are located in regions with high ICC values. However, other face-selective areas (e.g. pSTS and MFA), which are located anterior and medial to the occipital cortex, show weak genetic influence, consistent with the fact that the ICC values are low in those regions (compare Figure 2b and Figure 1a). Thus, with respect to the complete set of category-selective areas, one may conclude that the overall heritability of face-related activations is weaker. We have however emphasized in the Discussion that the genetic influence specifically in FFA and OFA is comparable to that in body and place areas.

A second concern regards the fact that all three domains appear to be genetically determined to some extent. Why would that be the case? Is it that functional domains that have been easy to guess by experimenters as possibly involving functional specialization and lead to very focal activation happen to also be genetically more determined than other functions? Or is it that, maybe, all parts of the brain are equally genetically determined, just that we have an easier time with the spatial resolution of the functional maps we can obtain and the inherent errors of cross-registration etc. revealing these genetic contributions in these domain specific areas? It would be great if the authors could run a simulation of

the latter possibility and generate an argument that this would not provide an explanation for their results. (This is more a request than a requirement.) I think this would be important for interpretation.

The reviewer has raised an interesting question: are face-, body-, and place-related activations genetically determined equally in all parts of visual cortex or are the genetic effects stronger in domain-specific areas? We did an analysis to specifically address this question (Figure 1 in the response letter). This figure was added to the manuscript as the new Supplementary Figure 1. In this analysis, we quantified the spatial overlap between the whole-cortex activation patterns in each twin pair. The spatial overlap was higher in MZ twins than DZ twins, especially when only vertices/patches with higher category specificity were included in the maps. This result suggests that the genetic effects are stronger in domain-specific areas.

Figure 1: Spatial overlap between category-selective activations in MZ and DZ twins. In each subject, face-, body-, and place-related activation maps were thresholded at different z values (z threshold: 0 to 8 with steps of 0.5), and the resulting maps were binarized. For each twin pair, the spatial overlap between the binarized maps was calculated using Dice coefficient. The overlap coefficients were then averaged across all MZ and DZ twin pairs.

It would be great if there were other functional specializations that do not show the effect. For example, can the authors analyze object-selective area LOC and show that it is not genetically determined?

In our initial plan, we wanted to test the genetic influence in the object-selective cortex as well. However, unlike other category-selective areas, we could not localize object-selective areas (including LOC) in a very specific way. As shown in Figure 2 in the response letter, we observed a widespread activation for objects in early visual cortex in addition to few scattered regions in high-level occipitotemporal cortex. We think that the following reasons may have contributed to these non-specific maps: A) The object stimuli in HCP are only tool images. For a robust localization of LOC, images from a wide range of object classes (not just tools) are typically used. B) In the literature, the classic contrast for localizing LOC has been 'objects vs. scrambled objects'. The scrambled stimuli have not been used in the tasks of HCP.

Figure 2: Object-selective activations in HCP. The object-selective voxels were defined as the top 1% of voxels which had the highest z values in the contrast of tool activity vs. average activity for all other categories, in the group-average maps. Similar criteria were used for defining face-, body-, and place-selective voxels. The maps are displayed on a flat patch of left and right hemispheres, and they are based on MSMSulc registration.

Relatedly, I am wondering what about other, higher-level functional specializations and whether they correlate. Given that subjects in this study were engaged in a number of different tasks, higher-order social cognition or speech processing could serve as a comparison. Alternatively areas thought to support general intelligence might serve as a point of comparison. Again, the point is to show whether there are discrete functions that might not show a genetic advantage.

These are all interesting suggestions that can be tested empirically using the large sample of twin subjects in the HCP database. A comprehensive evaluation of genetic influence in the high-level cognitive networks/areas (e.g. multiple-demand network, language areas, and areas involved in social cognition and emotion processing) can be considered in a subsequent study. Just to remind the reviewer, our results show heterogeneity of genetic influence in the category-selective areas. Some areas or subregions within some areas actually do not show any genetic effects.

Reviewer 2

The authors report the results of a large scale reanalysis of data available through HCP to test whether there is greater similarity in activity patterns and structural brain morphology between monozygotic twin pairs compared to dizygotic twin pairs. The principal finding is that there is a significant contribution of genetics to similarity in category-specific areas of the ventral stream.

This is an important contribution on an important topic. The issues highlighted below are primarily focused on the need for the theoretical contribution of the paper to be better situated in the setting of the prior literature. I also found it surprising that the authors did not use an 'unrelated' participant pair baseline: i.e., it seems that the study design begs for monozygotic, dizygotic and unrelated pairs.

We thank the reviewer for a positive assessment of our manuscript. Please see responses to your specific comments below.

1. The authors only point out midway through the paper that a prior study (Polk et al) found similar findings in terms of functional activity; that prior study was within far fewer subjects, but its not clear if

the authors expansion of experimental power is more than an incremental advance. The key issue that the authors need to specify what is theoretically novel about their contribution—my sense is that combination of analysis of functional activity and gyral folding and myelination frames a potentially important theoretical contribution that is hinted at but not fully developed.

We believe that our work has four key contributions (two technical and two theoretical; see below), and therefore it can be considered as an important advancement over the study by Polk et al.

- 1) The genetic influence in category-selective areas has been tested in a large sample of twin subjects (technical contribution). The study by Polk et al. was done in a much smaller sample size.
- 2) The contribution of genetic and environmental factors has been quantified using structural equation modeling (technical contribution). The study by Polk et al. was based on just correlation analysis.
- 3) Our study showed for the first time that the genetic influence was not homogeneous across the category-selective areas; only parts of these areas showed significant genetic effect (theoretical contribution).
- 4) Our study showed for the first time that the genetic influence had a link with the pattern of cortical curvature, cortical thickness, and cortical myelination in the category-selective areas (theoretical contribution).

We modified the first paragraph of Discussion to further emphasize these important points.

Please also see the comment by reviewer #3 regarding the novelty of our study.

2. Why did the authors not carry out the analyses for unrelated pairs (e.g., reshuffles on the twin data to make genetically and experientially unrelated pairs)? If there is an effect of genetics then it should be greater for dizygotic than unrelated pairs, right? This seems like an important validation and extension that is not tested even though the authors have the data.

Thank you for your nice suggestion. We did the requested analysis (Figure 3 in the response letter). This figure was added to the manuscript as the new Supplementary Figure 3. In this analysis, we compared the correlation of activation patterns between MZ twin pairs, DZ twin pairs, and unrelated pairs in face, body, and place areas. The correlation was higher in MZ twins than DZ twins, and in DZ twins than unrelated pairs, suggesting a strong genetic influence in category-selective areas. This result was also consistent with the graded genetic similarity of MZ and DZ twins (MZ twins share all their genes, while DZ twins share on average half of their genes).

Figure 3: Similarity of functional activation patterns in category-selective areas. Based on an analysis described in Figure 1b, the Pearson's r correlation was computed between the activation patterns in MZ twin pairs, DZ twin pairs, and unrelated pairs. Only face-, body-, and place-selective voxels (913 voxels for each category) were included in this analysis. Unrelated pairs were generated by shuffling subjects in the DZ group (similar results were obtained with shuffling subjects in the MZ group). In each category, the mean correlation varied significantly ($p < 0.0005$; one-way ANOVA) across the three groups, and all pairwise comparisons were significant ($p < 0.05$; Tukey's HSD Post-Hoc test), except for the comparison of DZ twins versus unrelated pairs in the face category ($p > 0.05$).

3. Large swaths of the prior literature are not acknowledged; this has the effect of weakening the contribution of the study rather than making it seem more novel. For instance, the authors reference a single study with congenitally blind subjects, but there is a lot more work in that space that has already made the argument that the broad organization by semantic category in the ventral stream is driven by innate constraints, and that those innate constraints pertain to the connectivity of the ventral stream. This argument about the role of innate connectivity constraints on shaping the organization of ventral stream was initially made by Caramazza and coauthors (see refs below) and subsequently picked up by Kanwisher and colleagues and others. Those arguments may also tie into the discussion in the GD about axonal tension and cortical folding.

We have now cited the paper by Mahon et al., 2009 in the section where we mentioned studies on congenitally blind subjects (sorry for forgetting to cite this highly relevant paper): However, using auditory stimuli representing different categories, recent studies have demonstrated a similarity in the functional organization of category-selective ventral temporal cortex in congenitally blind subjects and sighted controls – arguing that the development of category-selective map in visual cortex does not rely on visual input and visual experience (Mahon et al., 2009; van den Hurk et al., 2017).

We also cited the papers by Mahon and Caramazza, 2011 and Wang et al., 2017 in the Introduction. The antepenultimate sentence of the second paragraph in the Introduction: It has been proposed that the broad organization of ventral visual stream is driven by innate connectivity between regions that process semantic categories (Mahon and Caramazza, 2011; Wang et al., 2017).

The proposal by Mahon and Caramazza was also mentioned in the Discussion in the context of axonal tension and cortical folding: The link between genetics and cortical folding could then be explained by tension-based theory of cortical morphogenesis (Van Essen, 1997). According to this theory, cortical

connections between functionally-related areas are formed during the prenatal stage of cortical development (see also Mahon and Caramazza, 2011).

4. The framing of the paper rests on a false dichotomy: the dichotomy that is set up in the introduction is between ‘genes’ and ‘experience’. The dichotomy is false because there are not theories of innate constraints that do not require experience (i.e. development). Even something as prosaic as ‘growing a hand’, which is clearly genetic, requires an environment (i.e. experience). A little closer to home—if you assume human language results from a genetic endowment, that theory is not embarrassed by observations that children (tragically) raised without linguistic input do not develop normal language abilities. There is no zero sum between genetics and experience; the question is whether there is a contribution of genetics, quite independent of whether there is a contribution of experience. So the results from Arcaro et al, for instance, do not show that face specificity is not genetically constrained—just that experience is necessary (but not genetic story denies you need experience). More subtle engagement with these asymmetries would greatly strengthen the paper. An excellent example of this playing out in fact is the debate about ‘congenital’ prosopagnosia, and whether it points to genetic contributions to face specificity (see for instance, recent exhaustive review with commentaries, by Geskin and Behrmann, 2018).

We agree it is wrong to think of genes and environment as alternatives, and have revised the text to avoid this implication. The first sentence of the second paragraph in the Introduction: A prominent yet unresolved question concerns strength of genetic and environmental influences on the organization of category-selective areas. The penultimate sentence of the second paragraph in the Introduction: Finally, individuals with congenital prosopagnosia show lifelong difficulty in recognizing faces despite normal or almost normal exposure to faces, suggesting that the face recognition deficit in these individuals may have a genetic basis (Geskin and Behrmann, 2018).

5. The GD missed many opportunities to reinforce the theoretical contribution of the paper—the discussion of sparse versus distributed coding of faces and places, based on cortical thickness, seems incredibly speculative and ungrounded. I would encourage the authors to focus their discussion on what their paper is directly measuring and about—which is whether there are genetic contributions to category-specificity.

For instance, Buchel et al and then Striem-Amit et al have shown quite clearly that there is a visual word form area in the congenitally blind. Reading is not an evolutionarily old skill. So it cannot be that ‘reading’ is innate. And yet, visual experience is not needed for emergence of a reading area (in the blind it responds to braille). So clearly there is some non-visual constraint here—and probably (what else could it be?) it is the connectivity that is innate.

Would the authors predict greater similarity in the visual word form area in monozygotic than dizygotic twins? I think they would based on the data from the blind—so what would this mean for their argument?

Inspired by the reviewer’s comment, we added the following paragraph in the Discussion:

Previous fMRI studies have shown that the representation of visual field eccentricity (a low-level visual feature) extends into higher-tier category-selective cortex (Hasson et al., 2002). For instance, the face-selective FFA and the word-selective visual word form area (Cohen et al., 2000) contain a foveal representation, whereas the place-selective PPA contains a peripheral representation. The heritability of

category-specific representations could be, to some extent, related to the heritability of lower-level eccentricity representations. In the case of word area, evidence suggests that word-selective activations are heritable (Pinel et al., 2015), and that visual experience is not needed for the formation of this area (Büchel et al., 1998; Striem-Amit et al., 2012). Reading is not an evolutionarily old skill. Thus, it is unlikely that the word selectivity itself has a genetic basis. However, a more primitive form of representation in this area (namely, the foveal representation) could have emerged through innate mechanisms. In the case of face, body, and place areas, it remains an open question how much of the heritability in the activation patterns is related to the heritability of eccentricity representations.

References the authors may find useful (this is in no way exhaustive—I would encourage the authors to thoroughly look into the precedent of their claim for a genetic contribution to category-specificity in the ventral stream).

Büchel, C., Price, C., Friston, K. (1998). A multimodal language region in the ventral visual pathway. *Nature*, 394, 274–277.

Geskin, J., Behrmann, M. (2018). Congenital prosopagnosia without object agnosia? A literature review. *Cognitive Neuropsychology*, 35, 4-54.

Mahon, B.Z., Caramazza, A. (2011). What drives the organization of object knowledge in the brain? *Trends in Cognitive Sciences*, 15, 97-103.

Mahon, B.Z., Anzellotti, S., Schwarzbach, J., Zampini, M., Caramazza, A. (2009). Category-specific organization in the human brain does not require visual experience. *Neuron*, 63, 397-405.

Wang, X., He, C., Peelen, M.V., Zhong, S., Gong, G., Caramazza, A., Bi, Y. (2017). Domain selectivity in the parahippocampal gyrus is predicted by the same structural connectivity patterns in blind and sighted individuals. *Journal of Neuroscience*, 37, 4705-4716.

Striem-Amit, E., Cohen, L., Dehaene, S., Amedi, A. (2012). Reading with sounds: sensory substitution selectively activates the visual word form area in the blind. *Neuron*, 76, 640-652.

Saygin, Z.M., Osher, D.E., Norton, E.S., Youssoufian, D.A., Beach, S.D., Feather, J., Gaab, N., Gabrieli, J.D.E., Kanwisher, N. (2016). Connectivity precedes function in the development of the visual word form area. *Nature Neuroscience*, 19, 1250-1255.

These references have now been cited (see above).

Reviewer 3

Abbasi et al report an important set of findings on the heritability of category-selective responses in the cortex, using the Human Connectome Project database. They find that the spatial patterns of selectivity for faces, places, and bodies are more correlated between pairs of monozygotic (MZ) than dizygotic (DZ) twins, i.e., these spatial patterns of category selectivity are heritable. They further show that the most heritable voxels (“genetic voxels”) in the cortex within category-selective regions were located in voxels with higher curvature, and that cortex was thicker and more myelinated in genetic voxels of face areas, while it was thinner and less myelinated in genetic voxels of place areas. The paper is clearly written,

and the results will be of interest to a wide neuroscience audience. I had only one significant concern, and a few other questions and suggestions. Assuming the authors can address the significant concern (#1 below), I think the paper deserves to be published in Nature Communications.

We thank the reviewer (Nancy Kanwisher) for a positive assessment of our manuscript. We thoroughly addressed the major concern of the reviewer (please see below).

1. How much of the greater similarity in spatial patterns of functional response for MZ than DZ twins results simply from the overall greater similarity in the shape of the brain in MZ than DZ twin pairs? If DZ brains are more different in overall shape than MZ twins, then the cortical alignment is likely to be not as good for DZ as MZ twins, and as a result the functional correlations will likely be lower. The authors do show that some anatomical features of brains are heritable (like curvature and myelination; Figure 3). But the question here is whether there are heritability effects of the patterns of functional response over and above any anatomical differences in the mere shapes of the brains. One way to answer to this question would be to measure the similarity of the transformations required to register the two co-twin brains to the standard brain used in the study. If these transformations were more similar between two MZ than between two DZ twins, then the worry would remain that the apparently greater similarity in spatial patterns of functional response simply results from more consistent alignment to the standard brain for MZ than DZ twins. If the authors think this is not a reason for concern, clearly explain why to readers. (On a related note the authors should fix the colormap axis to the same upper threshold of 0.4 in both Figure 1 and 3 to aid the qualitative comparison between these plots).

We prepared three responses to the reviewer's insightful comment (How much of the greater similarity in spatial patterns of functional response for MZ than DZ twins results simply from the overall greater similarity in the shape of the brain in MZ than DZ twin pairs?):

First, we did the analysis requested by the reviewer. We measured the similarity of the transformations required to register the two co-twin brains to the standard brain used in our study. Using three different metrics, we confirmed the reviewer's conjecture: the registration-induced distortion maps were more similar between two MZ than between two DZ twins (Figure 4a in the response letter). However, an additional analysis showed that this similarity in the brain shape of MZ twins was a ubiquitous phenomenon that was present in both 'genetic' and 'non-genetic' category-selective cortex (Figure 4b in the response letter). Figure 4 was added to the manuscript as the new Supplementary Figure 8. Based on these analyses, we can conclude that the similarity of functional activation patterns in MZ twins is beyond the mere similarity in the brain shapes. Relatedly, Figure 1 in the response letter, which was provided for reviewer #1, shows that the spatial overlap between the activation patterns of one twin and the co-twin is higher in MZ twins than DZ twins, particularly when the most selective category-selective voxels are considered – again suggesting a link between genetic influence and functional properties of cortex.

Figure 4: Similarity of registration-induced distortion/deformation maps in MZ and DZ twins. During the HCP analysis pipeline, the surfaces of each subject in the native space were transformed into a standard mesh template (fs_lr) through a spherical registration. For this transformation, three types of distortion maps were obtained: (i) 3D displacement maps, displacement/shift in the 3D coordinates of each vertex, (ii) isotropic distortion maps, local change in the size of vertex areas in the triangular mesh, (iii) anisotropic distortion (strain) maps, local change in the aspect ratio of vertex areas in the triangular mesh. In both isotropic and anisotropic maps, distortion for a vertex area was calculated as $\log(\text{Area_Distorted}/\text{Area_Original})$. For each vertex, an average distortion was obtained by averaging

distortions across triangles sharing the vertex. (a) In each twin pair, the 3D displacement and areal distortion maps of one twin and the co-twin were correlated using Pearson correlation. All vertices from two hemispheres were included in this correlation analysis. Distribution of correlation coefficient values was plotted for all MZ and DZ twin pairs. (b) The correlation analysis was performed for genetic and non-genetic voxels within face, body, and place areas. Mean correlation was plotted for MZ and DZ twins. Mean correlation was always higher in MZ twins than DZ twins – though the difference was not significant in some comparisons. *: $p < 0.05$, **: $p < 0.005$, ***: $p < 0.0005$; pooled t-test, Bonferroni-corrected for 3 comparisons in panel b.

Second, we directly asked the reviewer's concern in the HCP mailing list. The question initiated an interesting discussion with the HCP team (Timothy Coalson, Mathew Glasser, and Michael Harms). Tim gave us a comprehensive answer. Our overall understanding from his answer was that the reviewer's

concern may not be too serious! His main points have been copy/pasted below. To see the complete thread of emails, please visit this link:

https://groups.google.com/a/humanconnectome.org/forum/m/?utm_medium=email&utm_source=footer#!topic/hcp-users/0QpH4QFKz7I

Tim's viewpoint:

"First, I expect it is established fact that MZ twins have more-similar anatomy than DZ twins, so the reviewer's suggested test doesn't seem to be in earnest. Since we don't have data that follows MZ anatomy with DZ functional consistency (or vice versa), the reviewer's hypothesis that the observations are completely explained by the registration consistency seems to be untestable. Furthermore, the thing that would actually need to be quantified to nail it down is the residual misalignment transform after registration (which is of course unknown, as otherwise we would just make a better registration in the first place), so evaluating similarity of transforms for the current registration isn't really the right approach anyway (as a hypothetical perfect alignment for every subject would likely still look more similar between MZ twins). Even with this magical knowledge about the ideal registration, you would still need a highly predictive model of registration quality on consistency of activations in order to say what the effect of the registrations you actually have would be.

However, shape differences are less of a problem for surface-based registration than for volume-based, and it seems like the reviewer may be thinking in terms of volume-based registration. If you happened to have another task that activated most of the same areas, but had less (or more) of a difference between MZ and DZ twins, that would imply that registration isn't the sole causal factor, and could place a bound on the total effect of registration (under the assumption that MZ twins won't ever be *less* similar on average than DZ). However, I wouldn't be surprised if most tasks either use different areas or have similar MZ vs DZ effects.

I'd like to again point out that if the distortion/transform is more similar across MZ pairs, that doesn't actually prove that the reviewer has a point - the reviewer was arguing that the remaining *lack* of perfect area-to-area registration would be more similar in the MZ case, on the presumption that anatomical shape has a substantial impact on the registration used, to such an extent that the registration imperfections could explain the entire functional result you found, by also being heritable (if each MZ pair starts with more-similar anatomy, and anatomy generally prevents the registration from fully aligning areas, then the residual non-registration will be in the direction of the original anatomy of those twins).

To put it another way, if the registration were perfect, such that it produced 100% overlap of areas all the time, there would be no heritability confound from the registration *quality* as the reviewer proposed, but the distortions and transforms of the registrations could still easily be heritable, which would basically only mean that the anatomical shape was heritable (because it has an influence on the initial native sphere generation) - thus what you are proposing to test may be expected to show a MZ vs DZ difference when there is no problem."

Third, we obtained the ICC maps (consistency of category-selective activations) in the MZ and DZ groups based on a multimodal 'MSMAIL' registration (Figure 5 in the response letter). These ICC maps were virtually similar to the ICC maps reported in the manuscript. Thus, our results were not dependent on the registration method used to align the subjects' functional data. As pointed out by Matt Glasser and

Tim Coalson in the HCP mailing list, the MSMAll registration tries to optimize the alignment of functional areas across subjects, and thereby there would be less worry about differences in the brain shape.

Figure 5: ICC maps in MZ and DZ twins based on MSMAll registration. The maps were obtained separately for face, body, and place categories.

The range of color scale bar in Figure 3b was changed to [-0.3 0.3] to be consistent with the range of color scale bar in Figure 1a.

2. What exactly does heritability mean?

Although I did not see a misinterpretation of the meaning of heritability, I think the paper would benefit from the addition of a sentence or two clarifying what exactly heritability does and does not mean. As others have written in detail (see for example, Block, *Cognition*. 1995 Aug; 56(2):99-128.), what we really want to know about the brain is genetic determination: that is, which properties of the brain are determined by genes, versus which result from experience. But that is not what heritability measures. Heritability measures how much of the variation between individuals is accounted for by genes, and how much by experience. Heritability can diverge importantly from genetic determination. For example, as Block notes in the reference above, having five fingers is clearly genetically determined. However, the heritability of finger number is almost zero, because most of the variation found in finger number in the population is most likely due to experience (in this case accidents), not genes. As another way in which heritability of a given characteristic does not at all mean that that characteristic is genetically determined, note that the genetic influence found in this study could exert its effects in all kinds of indirect ways, including via experience. As just one example, perhaps what is heritable is the subject's

eye fixation behavior on faces (and places and bodies). If so, the precise retinotopic location of different features during experience would be heritable, and perhaps with it the precise location of that selectivity in the ventral visual pathway. In this case the heritability of the cortical patterns reported here could reflect primarily an experiential effect in disguise, operating via a genetic effect on eye fixation behavior. That is just one of many possible examples of how a heritability effect can be mediated by experience. I would like to see the authors mention some of these subtleties in the paper that others will not take these results as indicating genetic determination of the category selective regions themselves.

In the manuscript, we used the term heritability in the context of genetic modeling. That is, how much of the variation in a phenotype (here, category-selective activations) is accounted for by genetic factors/components. A new sentence was added in the third paragraph of the Introduction to further clarify what we mean by the heritability: One important aspect of genetic influence is the heritability of individual differences, as addressed in classical twin studies.

3. The main question, analysis, and results from the paper are very similar to those from the Polk et al study from 2007, so one might worry about how novel the paper is. I would actually argue that the N is so much larger here, and the result so worth replicating, that the paper is nonetheless worthy of publication in Nature Communications. If our field is to get its act together we need to place more weight on solid results and replications than on novelty. Further, the relationship to cortical curvature is novel, as is the analysis of cortical thickness and tissue myelination within face and place areas.

We agree with the reviewer, and we are grateful for her support. The reviewer's points here have been used for responding to the reviewer #2's comment regarding the novelty of our study (see the list of the contributions of our work in our response to the reviewer).

Other Minor Points

There may not be room in this short format, and I do not at all insist, but it would also be nice to see the authors mention some other things:

i) Brief mention of the heritability of face recognition behavior, which several studies have reported:
<https://www.ncbi.nlm.nih.gov/pubmed/20176944>
<https://www.ncbi.nlm.nih.gov/pubmed/20060296>

We added a sentence in the Introduction and cited these studies. The last sentence of the second paragraph in the Introduction: Genetic determination of face-processing system has also been suggested by studies reporting the heritability of face recognition behavior (Zhu et al., 2010; Wilmer et al., 2010).

ii) How might the results of the current study be reconciled with the Arcaro et al findings? I think they can be, easily, but it might help to spell this out.

Our results demonstrated that the genetic effects in the whole network of face areas were relatively weak compared to the genetic effects in the other category-selective networks. This statement has been mentioned in the Discussion, and it appears to be consistent with the Arcaro et al findings.

Small point: If the medial face area (MFA) has been reported before, please include a citation.

Face-related activations in posterior cingulate cortex / precuneus were originally reported by the Haxby group (Haxby and Gobbini, 2011). These activations are part of an 'extended neural system' for face processing, and they are usually localized using the comparison of familiar/famous faces versus unfamiliar faces. However, in the group-average map of HCP, which was based on a large number of subjects and a better inter-subject alignment, they were also robustly localized in the comparison of faces versus other object categories. Since the posterior cingulate cortex is a large anatomical area comprising many functional subregions, we decided to use the term 'medial face area' – MFA for a subregion which is face-responsive. Such naming of cortical areas based on function (rather than underlying anatomy or gyrus/sulcus) has actually been promoted by the Kanwisher group, as they used the terms medial place area and occipital place area to refer to place-selective activations in retrosplenial cortex (retrosplenial complex) and transverse occipital sulcus, respectively. In the caption of Figure 2, we have now clarified this naming issue: The face-selective area in the posterior cingulate cortex (Haxby and Gobbini, 2011) was named MFA (medial face area) here.

Haxby, J.V. & Gobbini, M.I. (2011). Distributed neural systems for face perception. In A.J. Calder, G. Rhodes, M.H. Johnson & J.V. Haxby (Eds.), *The Oxford Handbook of Face Perception*. Oxford University Press, Oxford, pp. 93-110.

At the end, we would like to thank the three reviewers for their valuable comments and suggestions.

Reviewers' Comments:

Reviewer #1:

Remarks to the Author:

I thank the authors for responding to my concerns. I think that it is odd to draw a general conclusion about heritability of face selectivity in the beginning of the Discussion, "consistent with experience-dependent development of face patches", and to then present the results from FFA and OFA as a caveat. I find the response in the reviewer rebuttal much more convincing: "In the genetic analysis of category-selective areas in Figure 2, FFA and OFA show strong genetic influence. FFA and OFA are located in regions with high ICC values. However, other face-selective areas (e.g. pSTS and MFA), which are located anterior and medial to the occipital cortex, show weak genetic influence, consistent with the fact that the ICC values are low in those regions.." Some face areas show the genetic effects, others do not. Why not state the pattern found in this manuscript in the Discussion?

Reviewer #2:

Remarks to the Author:

The authors have responded carefully and thoroughly to the issues that I raised previously in the first round of review—this paper makes a substantial contribution to the field and will become a standard for how to do these types of large scale studies that unpack the genetic contributions to functional brain organization.

I have three comments below that can be addressed through textual revision and which I could classify as minor issues-

1. The claim that the organization of category-preferring regions is driven by genetic factors is not incompatible with a role for experience (we know, prior to running any experiments, that it can't be just genes, because experience is needed to fill in the content). The current ms is much more clear on this but there is still some unnecessary tension on this point in the introduction....ie the tension built up in the introduction still feels somewhat contrived and it is not needed to motivate the study. I think it would be more powerful frame the investigation more squarely as being able to really assess exactly the contribution of genetic factors to functional organization (while carefully parsing out the contribution of genetic factors to structural similarity).

2. Last paragraph of introduction—the prior study by Polk et al should (in my view) be acknowledged and referenced in the introduction because that is where the idea of comparing MZ and DZ twins is motivated, and that prior study did just that (and to reiterate the point from my first review—the current study goes so far beyond that prior study in scale, depth and sophistication of analyses, and thus theoretical confirmation, that I see no cost with respect to 'novelty' or 'conceptual advance' by fully acknowledging that prior study up front and summarizing the core findings in a sentence or two in the introduction). The current study makes a novel and substantial contribution even if the core logic is shared by that prior paper-

3. I found this logic to require a bit more unpacking: "Reading is not an evolutionarily old skill. Thus, it is unlikely that the word selectivity itself has a genetic basis." Just because reading is evolutionarily new does not mean that the specialization that undergirds the VWFA is not driven by genetic factors—it just means that the function the region was selected for was not reading. Compare—there may be a huge genetic contribution to being a great baseball player...even though 'playing baseball' was not the

substrate for evolution. For instance, as suggested by Martin (2006, *Neuron*, 50, 173-5) and Mahon and Caramazza (2011, as cited) it could well be that connectivity to language areas is what drives specialization for printed words...and that connectivity could well have been selected for (just not in the setting of reading per se). The authors may find the arguments in Dehaene and Cohen (2007, *Neuron*, 56, 384-98) on cultural recycling, as well as the neuroanatomical evidence about connectivity of the visual word area in Bouhali et al (2014; *J. Neurosci*, 34, 15402-15414) to language areas to be relevant to the further development of this point.

Reviewer #3:

Remarks to the Author:

The authors have submitted an unusually thorough and excellent response to the reviewer comments, and this important paper is now ready for publication.

Response Letter

Reviewer #1 (Remarks to the Author):

I thank the authors for responding to my concerns. I think that it is odd to draw a general conclusion about heritability of face selectivity in the beginning of the Discussion, “consistent with experience-dependent development of face patches”, and to then present the results from FFA and OFA as a caveat. I find the response in the reviewer rebuttal much more convincing: “In the genetic analysis of category-selective areas in Figure 2, FFA and OFA show strong genetic influence. FFA and OFA are located in regions with high ICC values. However, other face-selective areas (e.g. pSTS and MFA), which are located anterior and medial to the occipital cortex, show weak genetic influence, consistent with the fact that the ICC values are low in those regions.” Some face areas show the genetic effects, others do not. Why not state the pattern found in this manuscript in the Discussion?

We now revised the Discussion based on the reviewer’s comment.

Reviewer #2 (Remarks to the Author):

The authors have responded carefully and thoroughly to the issues that I raised previously in the first round of review—this paper makes a substantial contribution to the field and will become a standard for how to do these types of large scale studies that unpack the genetic contributions to functional brain organization.

I have three comments below that can be addressed through textual revision and which I could classify as minor issues.

1. The claim that the organization of category-preferring regions is driven by genetic factors is not incompatible with a role for experience (we know, prior to running any experiments, that it can’t be just genes, because experience is needed to fill in the content). The current ms is much more clear on this but there is still some unnecessary tension on this point in the introduction...ie the tension built up in the introduction still feels somewhat contrived and it is not needed to motivate the study. I think it would be more powerful to frame the investigation more squarely as being able to really assess exactly the contribution of genetic factors to functional organization (while carefully parsing out the contribution of genetic factors to structural similarity).

We now revised the Introduction based on the reviewer’s comment.

2. Last paragraph of introduction—the prior study by Polk et al should (in my view) be acknowledged and referenced in the introduction because that is where the idea of comparing MZ and DZ twins is motivated, and that prior study did just that (and to reiterate the point from my first review—the current study goes so far beyond that prior study in scale, depth and sophistication of analyses, and thus theoretical confirmation, that I see no cost with respect to ‘novelty’ or ‘conceptual advance’ by fully acknowledging that prior study up front and summarizing the core findings in a sentence or two in the introduction). The current study makes a novel and substantial contribution even if the core logic is shared by that prior paper.

We now cited the study by Polk et al in the Introduction, as requested by the reviewer.

3. I found this logic to require a bit more unpacking: “Reading is not an evolutionarily old skill. Thus, it is unlikely that the word selectivity itself has a genetic basis.” Just because reading is evolutionarily new does not mean that the specialization that undergirds the VWFA is not driven by genetic factors—it just means that the function the region was selected for was not reading. Compare—there may be a huge genetic contribution to being a great baseball player...even though ‘playing baseball’ was not the substrate for evolution. For instance, as suggested by Martin (2006, *Neuron*, 50, 173-175) and Mahon and Caramazza (2011, as cited) it could well be that connectivity to language areas is what drives specialization for printed words...and that connectivity could well have been selected for (just not in the setting of reading per se). The authors may find the arguments in Dehaene and Cohen (2007, *Neuron*, 56, 384-398) on cultural recycling, as well as the neuroanatomical evidence about connectivity of the visual word area in Bouhali et al (2014, *J. Neurosci*, 34, 15402-15414) to language areas to be relevant to the further development of this point.

In the same paragraph, we have mentioned that a more primitive form of representation in the visual word form area may have emerged through innate mechanisms. We have speculated that this representation might have been a foveal representation. However, it could also be a functional specialization related to language, as suggested by the reviewer. Since our paper is not about the heritability of activations in the visual word form area, we think that a lengthy discussion on this issue may not be necessary.

Reviewer #3 (Remarks to the Author):

The authors have submitted an unusually thorough and excellent response to the reviewer comments, and this important paper is now ready for publication.